# SuFP: Piecewise Bit Allocation Floating-Point for Robust Neural Network Quantization

**Geonwoo Ko**[*]                                                                 *geonwooko@kaist.ac.kr*
*Korea Advanced Institute of Science and Technology (KAIST)*

**Sungyeob Yoo**[*]                                                              *sungyeob.yoo@kaist.ac.kr*
*Korea Advanced Institute of Science and Technology (KAIST)*

**Seri Ham**                                                                         *seri1215@kaist.ac.kr*
*Korea Advanced Institute of Science and Technology (KAIST)*

**Seeyeon Kim**                                                                     *seeyakim@kaist.ac.kr*
*Korea Advanced Institute of Science and Technology (KAIST)*

**Minkyu Kim**                                                                 *minkyu.kim@krafton.com*
*KRAFTON*

**Joo-Young Kim**                                                           *jooyoung1203@kaist.ac.kr*
*Korea Advanced Institute of Science and Technology (KAIST)*

**Reviewed on OpenReview:** *https://openreview.net/forum?id=7M1adi1nfX*

## Abstract

The rapid growth in model size and computational demand of Deep Neural Networks (DNNs) has led to significant challenges in memory and compute efficiency, necessitating the adoption of lower bit-width data types to enhance hardware performance. Floating-point 8 (FP8) has emerged as a promising solution, supported by the latest AI processors, due to its potential for reducing memory usage and computational load. However, each application often requires its own optimal FP8 configuration to achieve high performance, resulting in inconsistent performance and increased hardware complexity. To address these limitations, we introduce Super Floating-Point (SuFP), an innovative data type that integrates various floating-point configurations into a single representation through a piecewise bit allocation. This approach enables SuFP to effectively capture both dense regions near zero and sparse regions with outliers, thereby minimizing quantization errors and ensuring full-precision floating-point performance across different models. Furthermore, SuFP's processing element design is optimized to reduce the hardware overhead. Our experimental results demonstrate the robustness and accuracy of SuFP over various neural networks in the vision and natural language processing domains. Remarkably, SuFP shows its superiority in large models such as large language model (Llama 2) and text-to-image generative model (Stable Diffusion v2). We also verify training feasibility on ResNet models and highlight the structural design of SuFP for general applicability.

## 1 Introduction

Deep Neural Networks (DNNs) have demonstrated exceptional performance across a wide range of applications, including Computer Vision (Deng et al., 2009a) and Natural Language Processing (NLP) (Chowdhary & Chowdhary, 2020). Moreover, DNNs are now excelling in state-of-the-art generative models, such as

---

[*]These authors contributed equally to this work.

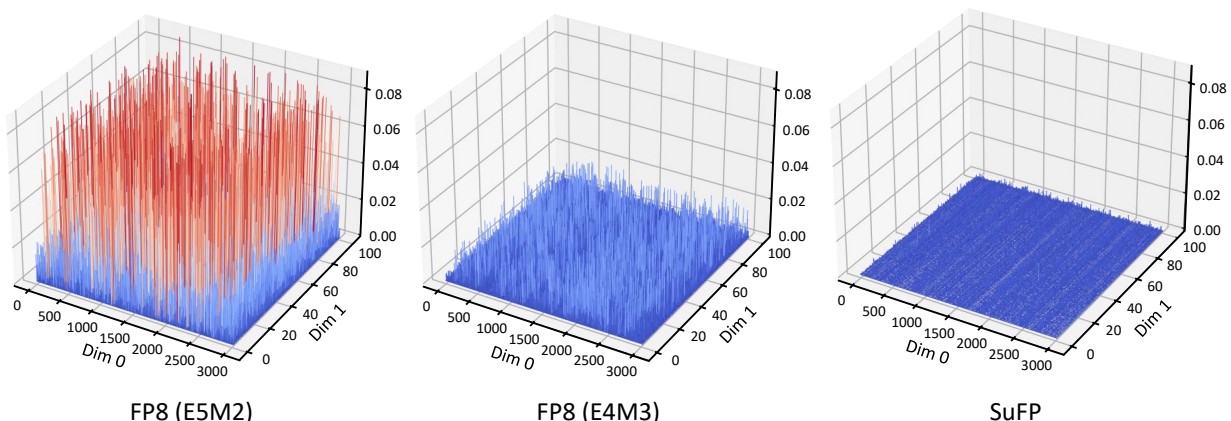

Figure 1: Comparison of the Mean Square Error (MSE) for FP8 formats and SuFP in a ViT-B/16 model activation layer. E5M2 and E4M3 are used as representative FP8 formats supported by NVIDIA GPUs. The SuFP format exhibits significantly lower MSE values compared to these IEEE-like FP8 formats.

text-to-image models (Rombach et al., 2022) and Large Language Models (LLMs) (Touvron et al., 2023). This extended applicability further elevates the standing and importance of DNNs within the broader AI landscape.

Recent progress in DNNs has largely come from scaling them up. This trend has led to exponential growth in model size and computational complexity, which in turn demands larger memory footprints and greater compute capacity (Gholami et al., 2021). This growth creates significant bottlenecks in every type of computational hardware, from servers to edge devices. Consequently, adopting lower bit-width data types is essential to enhance hardware performance. To address this demand, recent studies have focused on maintaining original model performance while improving its memory footprint and compute efficiency through low-bit operations.

Floating-point 8 (FP8) has emerged as a leading low-precision format for AI workloads and is already supported by recent NVIDIA GPUs such as H100 and Blackwell (NVIDIA, 2023; 2024). FP8 represents data through exponent and mantissa, similar to Floating-point 32 (FP32) and Floating-point 16 (FP16). However, due to its insufficient bit-width, FP8 struggles to accurately represent data distributions, as illustrated in Figure 1. The complexity and diversity of tensor distributions make it difficult to apply a single FP8 configuration across various applications. Each application requires an optimal configuration of exponent and mantissa bits (e.g., E5M2, E4M3, E3M4, and E2M5) tailored to its specific data distribution characteristics. Consequently, using a single FP8 configuration for multiple applications can lead to performance degradation. To address this, mixed FP8 configurations have been explored as one approach to optimize performance (Shen et al., 2023), but at the cost of extra algorithmic complexity and hardware overhead. These challenges highlight the need for innovative solutions to effectively leverage FP8 across various applications without compromising performance or increasing hardware complexity.

In this paper, we introduce Super Floating-Point (SuFP), a novel data type designed to overcome the limitations of FP8 by ensuring model performance, enhancing robustness, and optimizing hardware efficiency. SuFP employs a piecewise bit allocation strategy, integrating various floating-point configurations optimized for different regions of the data distribution into a single data type. This enables the effective representation of the data distribution across a wide range of models. Specifically, SuFP allocates different numbers of bits to capture both the dense near-zero region and the sparse region containing outliers, maximizing representation efficiency within a constrained bit-width. Figure 1 highlights the effectiveness of this approach by showing that our data type achieves lower Mean Square Error (MSE) across the entire tensor compared to various FP8 configurations. In this way, SuFP maintains low quantization errors across diverse data distributions in various models, ensuring broad applicability. Moreover, SuFP is highly optimized for hardware; its scaling bias, using power-of-two representation, only requires integer addition for scaling, whereas conventional FP8

necessitates FP32 multiplication. This enables coverage of a diverse dynamic range with negligible hardware overhead. Additionally, the tailored hardware for SuFP consists solely of an integer arithmetic unit and a shifter, allowing for a highly compact hardware configuration.

In summary, the contributions of SuFP are:

- **Ensuring Model Performance:** SuFP employs a piecewise bit allocation to integrate various floating-point configurations optimized for different regions of the data distribution into a single data type. This method ensures effective representation and minimizes quantization errors across diverse models, as demonstrated by the lower MSE compared to various FP8 configurations.

- **Enhancing Robustness:** By allocating different precisions to capture both the dense near-zero region and the sparse region containing outliers, SuFP adapts to a wide variety of data distributions. This flexibility ensures broad applicability and maintains low quantization errors across different models.

- **Optimizing Hardware Efficiency:** SuFP is highly optimized for hardware, using a power-of-two scaling bias that requires only integer addition for scaling. This approach, combined with a tailored hardware design consisting solely of an integer arithmetic unit and a shifter, enables coverage of a diverse dynamic range with negligible hardware overhead, resulting in a compact and highly efficient hardware configuration.

## 2 Related Works

**Integer data type.** Low-bit quantization in neural networks commonly uses integer data types due to their operational efficiency and straightforward implementation. With these data types, the full range of data values is uniformly mapped to the available integer levels. However, this method makes it difficult to accurately represent both dense data regions and sparse outliers, making it hard to maintain model accuracy.

To address these issues, various studies have proposed several methods. Xiao et al. (2023) introduces an extra scaling factor to handle outliers better. Similarly, Dettmers et al. (2022) employs higher precision for specific parts of the data to mitigate this issue. Due to the significance of handling quantization error, Wu et al. (2020) implements additional recovery methods, while Yao et al. (2022) incorporates knowledge distillation to improve accuracy. Additionally, Frantar et al. (2022) chooses to quantize only the weights to simplify the process and reduce errors. These various approaches highlight the ongoing challenges and complexities associated with low-bit integer quantization.

**Floating-point data type.** Floating-point data types present a promising alternative for low-bit quantization, especially as many neural network models exhibit Gaussian-like distributions in their weights and activations. Various studies have explored different aspects and methodologies for implementing low-bit floating-point quantization. Zhang et al. (2023) demonstrates that floating-point data types often outperform integer data types in activations. However, due to the complexity and diversity of tensor distributions, there is no universally superior format for all tensors or layers. This complexity and diversity indicate limitations in the general applicability of low-bit floating-point formats. In response to these limitations, alternative approaches such as the posit number system and EFloat have been explored. Gustafson & Yonemoto (2017) and Langroudi et al. (2020) propose the posit number system which offers higher accuracy and dynamic range compared to traditional floating-point formats. However the posit number system requires more complex hardware implementations. Our analysis additionally reveals that the posit number system is ill-suited to represent the tensor distribution of various DNN models. We discuss the details of these observations in the Appendix D. Similarly, Bordawekar et al. (2021) introduces EFloat, an entropy-coded format designed to reduce memory usage while maintaining precision; however it requires additional complex decoding hardware. To further address the diverse needs of low-bit quantization, Sun et al. (2019) suggests that mixed configurations of floating-point formats can provide robust accuracy improvements. Such insights, as highlighted in Kuzmin et al. (2022), may lead to hardware overheads since they do not take hardware efficiency into account. Furthermore, Micikevicius et al. (2022) and Wu et al. (2023) suggest

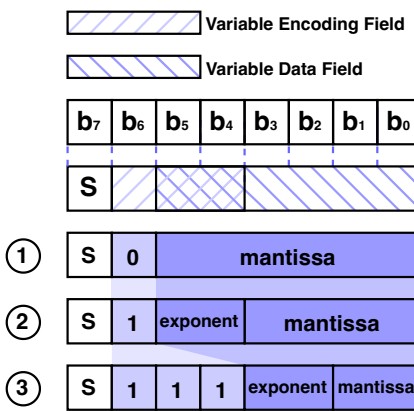

Figure 2: Visual representation of multi-region piece-wise bit allocation of SuFP.

**Algorithm 1:** SuFP Decoding Algorithm

**Input** : A Binary 8-bit Data $[b_7, b_6, \ldots b_0]$, scaling bias
**Output:** SuFP Quantized Data

1   $s_{sufp} \leftarrow b_7$
2   $bias \leftarrow Scaling\ Bias$
3   **if** $b_6 == 0$ **then**
4      $e_{sufp} \leftarrow -(100_2) \cdots \text{①}$
5      $m_{sufp} \leftarrow b_5 b_4 b_3 b_2 b_1 b_0 \cdots \text{①}$
6   **else**
7      **if** $b_5 b_4 \neq 11_2$ **then**
8          $e_{sufp} \leftarrow -(10_2) + b_5 b_4 \cdots \text{②}$
9          $m_{sufp} \leftarrow 1 b_3 b_2 b_1 b_0 \cdots \text{②}$
10      **else**
11          $e_{sufp} \leftarrow 11_2 + b_3 b_2 \cdots \text{③}$
12          $m_{sufp} \leftarrow 1 b_1 b_0 \cdots \text{③}$
13   **return** $(-1)^{s_{sufp}} \cdot m_{sufp} \cdot 2^{e_{sufp}+bias}$

that achieving high performance with low-bit floating-point data types often requires additional fine-tuning or the use of quantization error compensation schemes, which can introduce further complexity.

A common observation from these studies is that floating-point data types offer a more versatile approach to addressing diverse data distributions compared to integer data types. This versatility arises from their ability to adjust the exponent and mantissa bits to handle varying data ranges and resolutions. However, if the bit allocation between the exponent and mantissa is not optimized for each data distribution, it can negatively impact the overall accuracy and efficiency of the models.

**Task-specific quantization.** Task-specific quantization strategies aim to tailor quantization schemes to the specific characteristics of each model or task. Yuan et al. (2022) proposes twin-uniform quantization with Hessian-guided scaling to achieve near-lossless 8-bit performance on Vision Transformers. Wang et al. (2024) adopts distribution-aware activation quantization and structural risk minimization-based timestep selection to preserve image generation quality in diffusion models at 6-bit. In addition to post-training methods, recent studies have explored the application of low-bit precision during model training. Peng et al. (2023) introduces an automatic mixed-precision framework that extensively applies FP8 to most computation and storage paths for large-scale language models. However, task-specific quantization approaches often rely on customized functions and calibration procedures, which limiting their applicability to diverse models and tasks.

Recognizing these limitations, our work introduces SuFP, which overcomes the limitations of previous studies by optimizing bit allocations and utilizing variable encoding fields to ensure full-precision model performance, enhance robustness, and optimize hardware efficiency.

## 3   Super Floating-Point (SuFP)

The key idea of SuFP is multi-region piecewise bit allocation, which integrates various floating-point configurations optimized for different regions within data distribution into a single data type. This scheme allows SuFP to accurately represent both dense regions near zero and sparse regions with outliers, ensuring a wide dynamic range with efficient bit utilization.

**Multi-Region Piecewise Bit Allocation.** SuFP data type comprises the sign bit (MSB), an encoding field, and a data field. SuFP can present three different data representations of exponent-and-mantissa combinations based on the encoding field, as shown in Figure 2, and the data field is interpreted according to each representation.

The overall decoding process is described in Algorithm 1. Once the representation is determined, exponent ($e_{sufp}$) and mantissa ($m_{sufp}$) are determined based on the data field. Meanwhile, to achieve different

precisions, each representation has a distinct exponent baseline. The exponent baselines for representations ① , ② , and ③ are $-4$, $-2$, and 3, respectively.

There are other features that influence the precision of each representation. The representation ① is designed to express numbers close to zero with high granularity by using the entire data field for the mantissa. Notably, by setting $b_5$ and $b_4$ to $00_2$, it achieves the representation of subnormal numbers in the IEEE floating-point standard. On the other hand, both the representation ② and ③ divide the data field into exponent and mantissa sections. Specifically, the representation ③ aims to capture a broader range of numbers, including outliers, even with its fewer mantissa bits compared to representation ② . In contrast, the representation ② focuses on numbers within an intermediate range between representation ① and ③ . It offers finer precision due to its wider bit-width mantissa, allowing for an optimal expression of numbers between the main body and outliers.

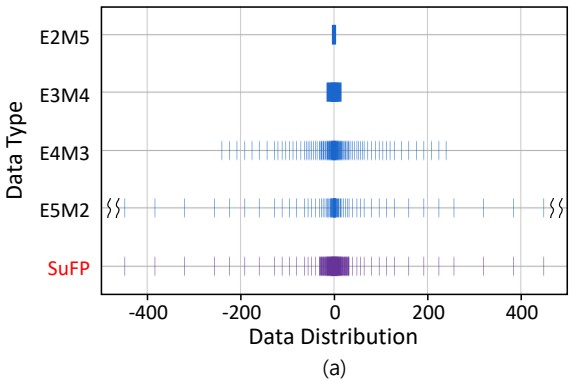
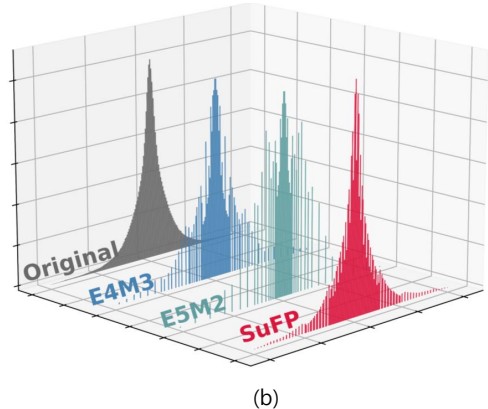

(a)                              (b)

Figure 3: (a) Illustrates the data distribution according to each FP8 configuration applying the scaling bias and (b) compares the quantized tensor distributions of the representative FP8 formats, E5M2 and E4M3, with SuFP. As shown in (b), SuFP data distribution most accurately represents the original tensor.

In summary, the encoding field is determined by comparing the exponent value of a given number against predefined threshold values. These thresholds segment the data distribution into regions, each represented by one of the three encoding schemes. Thus, numbers closer to zero are allocated more mantissa bits, while larger outlier values receive additional exponent bits.

Building on these specifics, each representation offers different levels of precision within its respective range, effectively covering a wide spectrum of numbers. For example, numbers in the range of 0 to 4, which belong to the main body, are captured with a granularity of $2^{-4}$. On the other hand, outliers in the range of 256 to 448 are captured with a granularity of $2^6$. Due to the implementation of the variable encoding field and variable data field, SuFP is capable of effectively representing both sparse and dense regions. This capability is illustrated in Figure 3, which demonstrates how SuFP follows data distributions compared to other FP8 configurations. For a detailed analysis of multi-region piecewise bit allocation, please refer to Appendix A and Appendix B.

**Computation with SuFP.** Before beginning the computation, we need a bias-selecting process as detailed in Appendix C. A scaling bias, which is the value added to the exponent baseline to determine the actual exponent value, allows for a more diverse range of exponent values than what can be originally represented with the limited exponent bit-width. Once the scaling bias for each tensor is predetermined, a tensor is quantized as SuFP without the need to search for the optimal scaling bias at runtime. This enables in-situ quantization for weight and activation tensors. Real-time quantization of the activation tensor is possible due to the highly consistent distribution of activation data during inference, regardless of input variations.

The inference process of SuFP consists of two main steps: decoding and computation. The decoding step is straightforward and involves first extracting the encoding field from the SuFP representation. Depending

on this encoding, the corresponding bit fields for exponent and mantissa are isolated. The exponent value is computed by adding the extracted exponent bits to the baseline exponent and the predetermined scaling bias.

Following the decoding step, the SuFP Arithmetic Logic Unit (ALU) is utilized to execute multiplications on mantissa and additions on exponents, as described in Equation (1) and (2). The largest value from the exponent addition results is identified. To align all the results with the maximum exponent value, the results of the mantissa multiplications are right-shifted, as shown in Equation (3). The aligned mantissa is then processed through an adder tree to compute the partial sum. Finally, after all computations on the two tensors are completed, the scaling bias is applied, as described in Equation (4). Unlike conventional FP8, which requires an FP32 multiplication unit for scaling, our SuFP only necessitates a simple integer adder since the scaling bias is a power of two. This difference emphasizes the hardware efficiency of our approach.

$$(\boldsymbol{w}_i \cdot 2^{\text{bias}_W}) \cdot (\boldsymbol{a}_i \cdot 2^{\text{bias}_A}) = \sum_{j=0}^{n-1} (-1)^{s_{w,j}} m_{w,j} 2^{e_{w,j}+bias_W} \cdot (-1)^{s_{a,j}} m_{a,j} 2^{e_{a,j}+bias_A} \tag{1}$$

$$= \sum_{j=0}^{n-1} (-1)^{s_{w,j}\oplus s_{a,j}} m_{w,j} m_{a,j} \cdot 2^{e_{w,j}+e_{a,j}} \cdot 2^{bias_W+bias_A} \tag{2}$$

$$(m_{w,j} m_{a,j})' = (m_{w,j} m_{a,j}) >> e_{max}, \ e_{max} = \max_{0 \le i \le n-1} (e_{w,i} + e_{a,i}) \tag{3}$$

$$(\boldsymbol{w}_i \cdot 2^{\text{bias}_W}) \cdot (\boldsymbol{a}_i \cdot 2^{\text{bias}_A}) \approx 2^{bias_W+bias_A+e_{max}} \cdot \sum_{j=0}^{n-1} (-1)^{s_{w,j}\oplus s_{a,j}} (m_{w,j} m_{a,j}) \tag{4}$$

**Hardware Implementation for SuFP.** Figure 4 shows the architecture of the proposed SuFP PE, designed to process 16 parallel input data streams. While the FP8 data types use floating-point-based PE, SuFP PE is composed mainly of integer operation ALUs and shifters, requiring fewer hardware resources. The SuFP PE can be readily integrated into any architecture, with the systolic array being a prime example. For a detailed explanation of the systolic array, please refer to Appendix G.

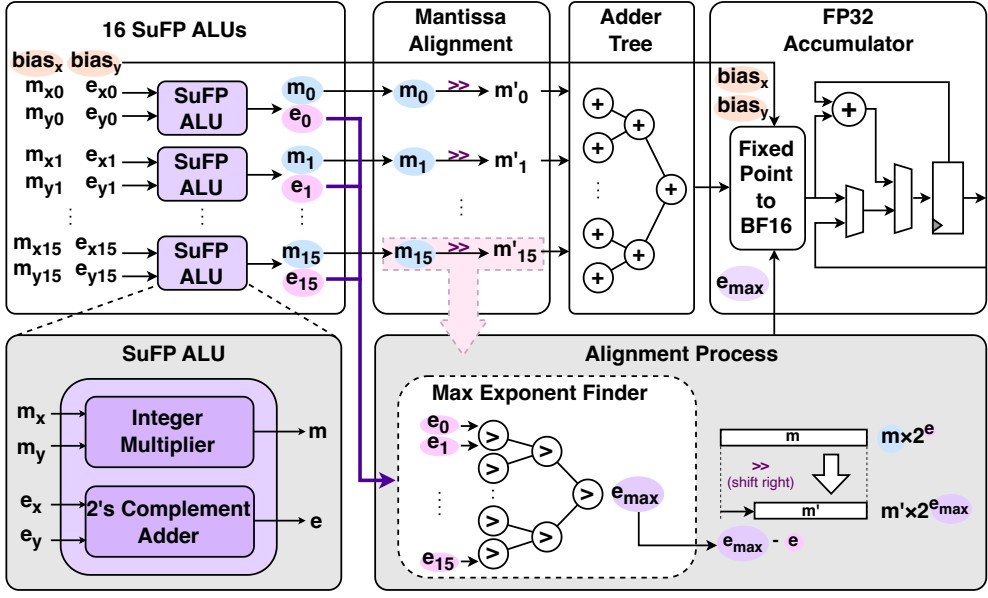

Figure 4: Proposed processing element architecture for SuFP.

## 4 Experiments

This section evaluates the proposed SuFP. We comprehensively assess SuFP's performance by quantizing both weights and activations across various models, including vision, language, and generative tasks. By comparing Quantization Signal-to-Noise Ratio (QSNR) (Darvish Rouhani et al., 2023) and hardware efficiency, we highlight the superior performance of SuFP over conventional FP8 configurations. Quantitative results show that SuFP achieves high accuracy across different models with minimal performance degradation, while qualitative results demonstrate high fidelity in image generation tasks. Additionally, we validate the applicability of SuFP in training neural networks, demonstrating its stability and precision with minimal performance drop.

### 4.1 Baselines and Experimental Setup

We implement SuFP using PyTorch with HuggingFace transformer and TorchVision libraries. We adopt post-training quantization (PTQ) for evaluating the effectiveness of our approach across various pre-trained models. For computer vision tasks, we benchmark our method on the ResNet18, ResNet50 (He et al., 2016), Vision Transformer (ViT) (Dosovitskiy et al., 2020), and EfficientNet-v2 (Tan & Le, 2021) models with the ImageNet dataset (Deng et al., 2009a). For natural language tasks, we benchmark our method using the BERT-base model (Devlin et al., 2018) on datasets such as MRPC, CoLA (Warstadt et al., 2018), and SQuAD 2.0 (Rajpurkar et al., 2018). For text-to-image generative tasks, we benchmark our approach using the Stable Diffusion v2 (Rombach et al., 2021) on the COCO dataset (Lin et al., 2014). For LLMs, we benchmark our method using Llama 2 model Touvron et al. (2023) on MMLU. We compare the performance of the proposed SuFP with the baseline data types, including FP32, FP16, Brain Floating-Point 16 (BF16), and different configurations of FP8.

Training experiments (Section 4.4) are conducted by emulating representable values of SuFP. We focus on input tensors of computation-intensive operations, such as convolutions or matrix multiplications, which contain weight, activation, and gradient tensors. We consider FP16 as a baseline and use the same hyperparameters from the baseline for SuFP. We train image classifier using ResNet-18 (He et al., 2016) and ResNet-50 (He et al., 2016) backbone on CIFAR-10, CIFAR-100 (Krizhevsky et al., 2009), ImageNet-100 (Deng et al., 2009b) and Tiny-ImageNet (Le & Yang, 2015) datasets. We train each classifiers three times with different seeds and report the mean of final accuracy.

Additionally, we use SystemVerilog to implement the SuFP PE and various configurations of FP8. All designs are synthesized using the Synopsys Design Compiler, optimized for the 28nm CMOS technology, and set to operate at 1 GHz clock frequency. In addition, we evaluate the power estimation of each PE with internal power, switching power, and leakage power. The implementation details of the previously mentioned PEs are elaborated in Appendix F.

### 4.2 Comparison of QSNR and Hardware Efficiency

In this section, we present an analysis of QSNR and hardware efficiency for SuFP and various FP8 configurations. QSNR is used to measure the numerical fidelity of various quantization schemes, with higher QSNR values demonstrating that the quantized values better follow the original values and their distribution (Darvish Rouhani et al., 2023). Hardware efficiency is expressed as the product of throughput and the number of operations, divided by the product of area and power consumption. This metric indicates how efficiently a system can perform a large number of operations within a given area and power budget. As shown in Figure 5, while conventional FP8 configurations face a trade-off between hardware efficiency and QSNR, SuFP overcomes this trade-off and performs highly in both aspects. The results show that SuFP consistently achieves high QSNR values and maintains advantages in hardware efficiency across various models and tasks, including vision, language, and generative applications. Specifically, SuFP shows reliable performance in both weight and activation quantization, indicating its robustness and versatility.

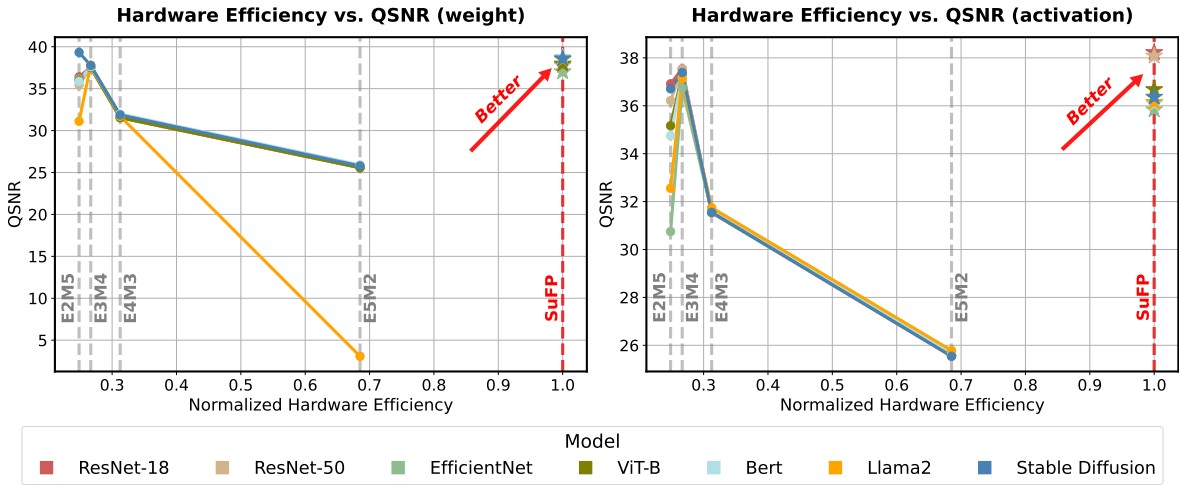

Figure 5: Comparison of hardware efficiency and QSNR for weights (left) and activations (right) across various models.

## 4.3 Inference Performance Analysis of Various Models

To highlight the robustness of SuFP across various domains, we evaluate the impact on the performance of our SuFP on various models. We extend our experiments to models in the vision, language, and generative tasks areas and compare the performance of SuFP with that of the standard formats like FP32, FP16, BF16, and different variations of FP8.

Table 1: Comparison of inference performance among various models with SuFP and other data types. For vision tasks, the models used in our experiments include ResNet (RN), EfficientNet (Eff.Net), and ViT-B/16, evaluated on the ImageNet benchmark for computer vision. For the NLP tasks, we use a BERT-base model, evaluated on MRPC, CoLA, and SQuAD 2.0 benchmark. Bold denotes the best result among 8-bit representation.

| Method | # bits | Vision | | | | NLP | | |
|--------|--------|--------|--------|---------|----------|--------|--------|-----------|
| | | RN18 | RN50 | Eff.Net | ViT-B/16 | MRPC | CoLA | SQuAD 2.0 |
| FP32 | 32 | 69.76 | 76.13 | 81.31 | 81.07 | 0.8307 | 0.5678 | 78.87 |
| BF16 | 16 | 69.80 | 76.11 | 81.31 | 81.04 | 0.8307 | 0.5635 | 78.86 |
| E5M2 | 8 | 67.04 | 72.96 | 78.59 | 80.42 | **0.8388** | 0.5793 | 78.55 |
| E4M3 | 8 | 68.82 | 75.32 | 80.43 | 80.84 | 0.8319 | 0.5566 | 78.56 |
| E3M4 | 8 | 69.55 | 75.83 | 81.15 | 80.99 | 0.8319 | 0.5604 | 78.70 |
| E2M5 | 8 | 69.70 | 76.00 | 77.86 | 80.88 | 0.8157 | 0.5511 | 78.46 |
| **SuFP** | **8** | **69.81** | **76.17** | **81.26** | **81.07** | 0.8371 | **0.5866** | **78.95** |

### 4.3.1 Quantitative Results

Table 1 compares the performance in the vision and NLP domains. For vision tasks, we evaluate ResNet-18, ResNet-50, EfficientNet-v2, and ViT-B/16 models using the top-1 accuracy metric. According to the experiments, SuFP results in a negligible accuracy decrease of less than 0.06% across all vision models, indicating that SuFP is comparable to full-precision in vision tasks. Moreover, SuFP outperforms conventional FP8 representations in terms of accuracy. For instance, SuFP achieves 81.26% accuracy for EfficientNet, only 0.06% below the original accuracy of 81.31%. In contrast, E3M4, the highest-performing conventional FP8 representation in this scenario, exhibits an accuracy of 81.15%, representing a 0.20% decrease compared to full-precision.

For NLP task, we evaluate the BERT-base model on the MRPC, CoLA, and SQuAD 2.0 datasets using accuracy, MCC, and F1-Score as the performance metrics, respectively. Based on the experimental results, SuFP demonstrates notably better results across various benchmarks for the BERT-base model compared to FP32. In addition, SuFP performs better than FP8s in most scenarios. These results indicate that SuFP delivers outstanding performance compared to FP8s with the same bit-width.

Table 2: Comparison of performance of Llama 2-7b with SuFP and other data types. Bold denotes the best result among 8-bit representation.

| Method | # bits | STEM | Humanities | Social Sciences | Other | Average |
|--------|--------|------|------------|-----------------|-------|---------|
| FP16 | 16 | 0.369 | 0.433 | 0.518 | 0.525 | 0.459 |
| E5M2 | 8 | 0.214 | 0.242 | 0.217 | 0.238 | 0.229 |
| E4M3 | 8 | 0.363 | 0.421 | 0.509 | **0.517** | 0.450 |
| E3M4 | 8 | 0.338 | 0.361 | 0.454 | 0.465 | 0.401 |
| E2M5 | 8 | 0.274 | 0.249 | 0.269 | 0.276 | 0.266 |
| **SuFP** | 8 | **0.370** | **0.426** | **0.516** | 0.516 | **0.455** |

Table 9 shows the performance of SuFP on LLM. We evaluate the Llama 2 model on the MMLU benchmark. Based on the experimental results, SuFP matches the baseline FP16 for this task, with performance decreasing only slightly by 0.87% from 0.459 to 0.455 on average. Notably, the fact that SuFP achieved high performance with just 8 bit-width quantization through direct adoption without fine-tuning is highly meaningful.

Based on the overall results, the optimal configuration for each model within conventional FP8 representations is inconsistent. For example, in the case of RN18, the E2M5 configuration achieves the best accuracy among conventional FP8 configurations, whereas in the NLP domain, the E5M2 configuration achieves the highest accuracy on the CoLA benchmark. In the LLM domain, the E4M3 configuration performs the best. This inconsistency makes it difficult to rely on a single FP8 representation to consistently achieve high performance across various models, highlighting the superior performance and versatility of SuFP.

### 4.3.2 Qualitative Results

We evaluate the performance of text-to-image generation model to demonstrate the effectiveness of the SuFP. As shown in Figure 6a, SuFP produces images with a quality similar to FP32, surpassing the conventional FP8 representations, such as E4M3, E3M4, and E2M5. This indicates that our SuFP has minimal information loss compared to these conventional methods. Additional samples of images generated by SuFP are included in Appendix H. Moreover, as shown in Figure 6b, unlike the conventional FP8 approaches, SuFP achieves details at the level of FP32, resulting in more realistic images. In particular, SuFP achieves an FID scroe of 25.6262 in our experiment using the COCO dataset, showing improved performance over FP32 recording an FID score of 27.0643. These results reinforce the reliability of SuFP and demonstrate its potential as an efficient alternative to FP32 for high-quality image generation.

### 4.4 Training

We further validate the broad applicability of SuFP, especially in training neural network scenarios, which typically requires precise approximation compared to inference with pre-trained models (Wang et al., 2018). Figure 7 visualizes the training process in terms of training loss. SuFP consistently shows stabilized training without severe fluctuation in the training loss curves.Table 3, 4, 5, 6 presents the final accuracy of each classifier. Classifiers trained with SuFP show a negligible drop in performance, at most 0.85%, which demonstrates stability of SuFP in neural network training.

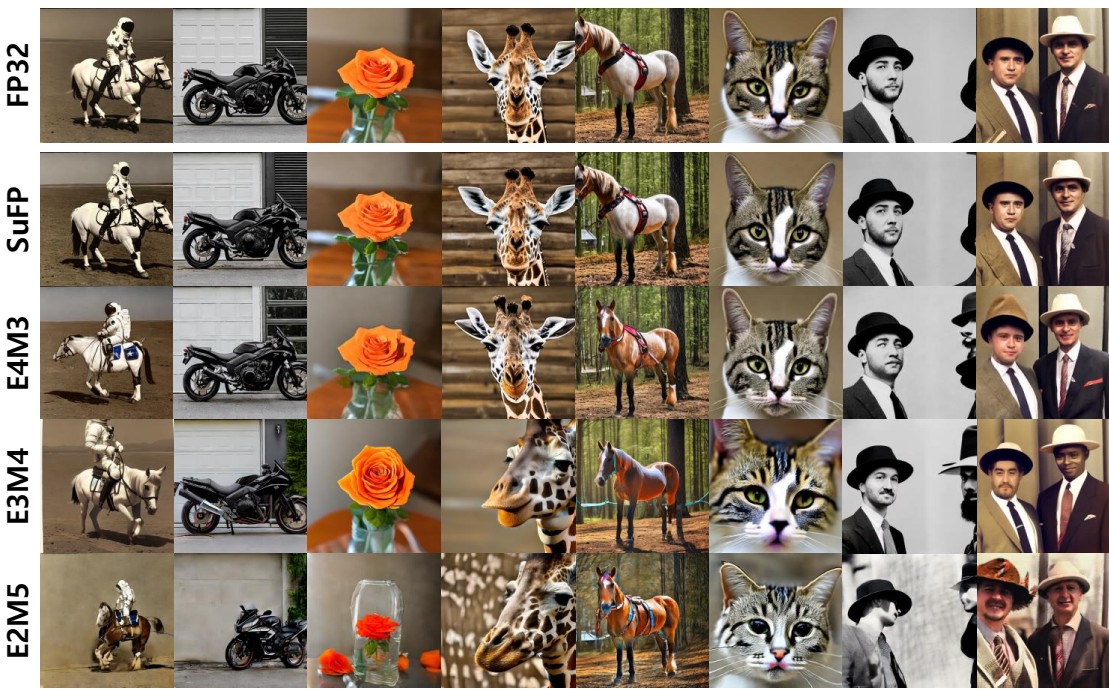

(a)

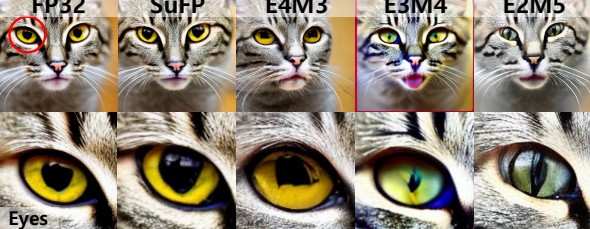

*"An old man in a suit and tie is staring."*   *"A cat that is staring at the camera."*

(b)

Figure 6: (a) Comparison of sample images and (b) details of images generated by the Stable Diffusion model v2 on the COCO dataset using FP32, SuFP, and FP8 representations (E4M3, E3M4, and E2M5). Note that E5M2 is excluded from the comparison because it fails to generate images. SuFP follows the original FP32 images more closely than other FP8 representations, capturing not only the overall composition but also subtle textures and fine-grained details, thereby demonstrating higher fidelity in quantization.

Table 3: ResNet18 Classification results on CIFAR-10 and CIFAR-100 dataset.

| Method | CIFAR-10 | | CIFAR-100 | |
|---|---|---|---|---|
| | Top-1 | Top-5 | Top-1 | Top-5 |
| FP16 | **94.87** | **99.84** | **76.95** | **93.61** |
| SuFP | 94.14 | 99.81 | 76.40 | 93.27 |

Table 4: ResNet50 Classification results on CIFAR-10 and CIFAR-100 dataset.

| Method | CIFAR-10 | | CIFAR-100 | |
|---|---|---|---|---|
| | Top-1 | Top-5 | Top-1 | Top-5 |
| FP16 | **95.37** | **99.92** | 78.29 | 94.57 |
| SuFP | 94.83 | 99.84 | **78.65** | **94.76** |

Table 5: ResNet18 and ResNet50 Classification results on ImageNet-100 dataset.

| Method | ResNet18 | | ResNet50 | |
|---|---|---|---|---|
| | Top-1 | Top-5 | Top-1 | Top-5 |
| FP16 | **81.98** | **95.50** | **84.54** | **96.48** |
| SuFP | 81.41 | 95.43 | 83.69 | 96.29 |

Table 6: ResNet18 and ResNet50 Classification results on Tiny-ImageNet dataset.

| Method | ResNet18 | | ResNet50 | |
|---|---|---|---|---|
| | Top-1 | Top-5 | Top-1 | Top-5 |
| FP16 | **51.38** | 75.98 | 54.51 | 78.51 |
| SuFP | 51.26 | **76.13** | **54.79** | **78.81** |

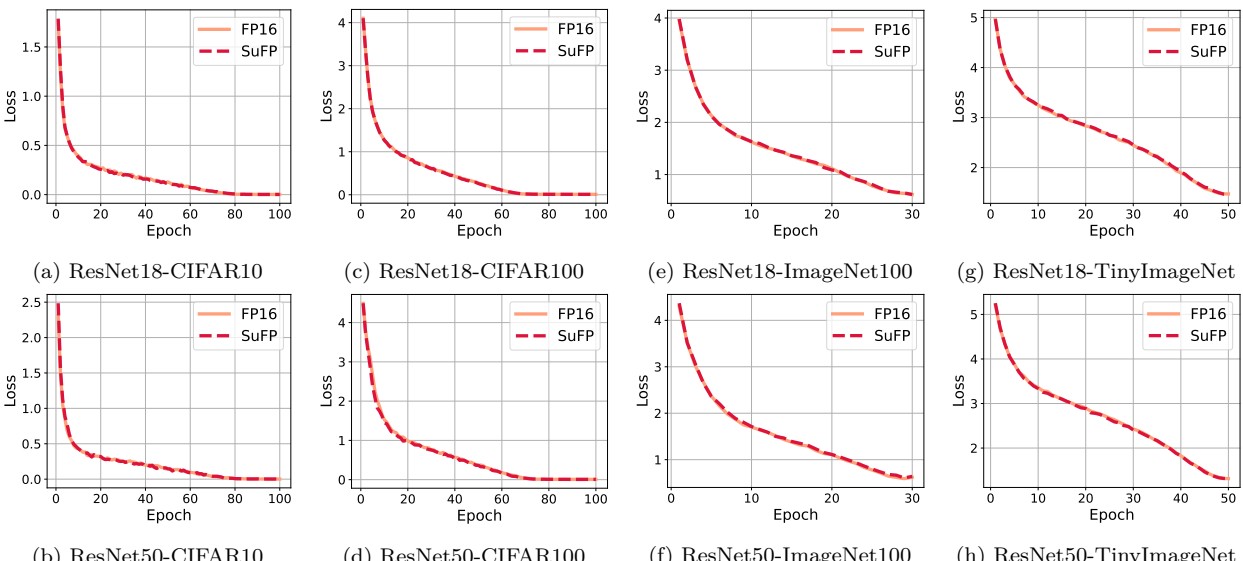

(a) ResNet18-CIFAR10  (c) ResNet18-CIFAR100  (e) ResNet18-ImageNet100  (g) ResNet18-TinyImageNet

(b) ResNet50-CIFAR10  (d) ResNet50-CIFAR100  (f) ResNet50-ImageNet100  (h) ResNet50-TinyImageNet

Figure 7: Visualization of training dynamics through training loss.

## 5    Conclusions

This paper introduces the Super Floating-Point (SuFP), designed to address the challenges of large and complex DNNs. SuFP optimizes precision in each data region by employing a multi-region piecewise bit allocation. By integrating various floating-point configurations into a single representation, SuFP effectively captures data in both dense near-zero regions and outlier regions, adapting to diverse data distributions. This approach overcomes the limitations of FP8, which often requires different configurations for each application, leading to inconsistent performance and increased hardware complexity. The tailored SuFP PE utilizes only integer units and shifters, ensuring a compact and optimized hardware implementation. Through this innovative method, SuFP demonstrates significant potential in enhancing the performance and efficiency of DNN computations, achieving accuracy comparable to FP32 and FP16. Consequently, SuFP is a compelling solution for the robust performance of overall AI applications.

**Limitation and Future work**   The quantization method proposed in this paper utilizes a static quantization technique that employs a power-of-two scaling bias. Therefore, applying a method that dynamically calculates precise floating-point scaling factors for tensors generated at runtime could potentially enhance performance. Nevertheless, our proposed method achieves near high-precision floating-point performance with a hardware-friendly scaling bias. Research into using more precise scaling factors for quantization remains an area for future investigation.

## Acknowledgements

This work was supported by IITP grant funded by the Korea government (MSIT) (No. 2022-0-01036, Development of Ultra-Performance PIM Processor SoC with PFLOPS-Performance and GByte-Memory and IITP-2025-RS-2023-00256472, Graduate School of Artificial Intelligence Semiconductor).

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

## Appendix A  Effectiveness of Multi-Region Piecewise bit allocation

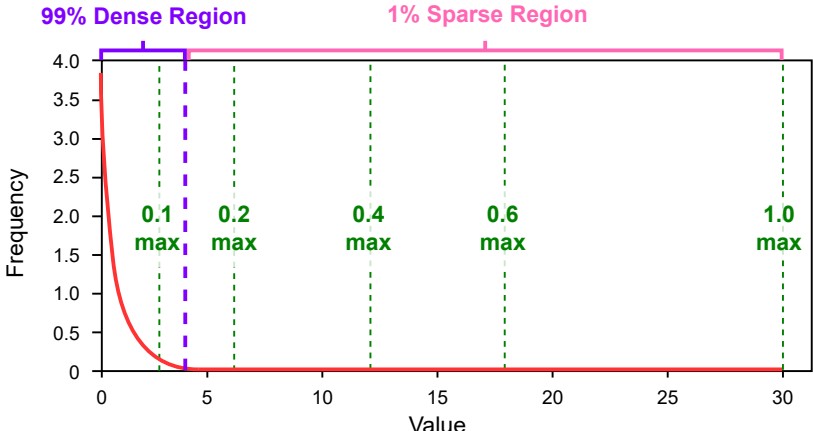

Figure 8: Real data distribution and piecewise bit allocation boundaries.

In this section, we discuss the effect of multi-region piecewise bit allocation applied to SuFP. As shown in Figure 8, the real data distribution can be divided into a dense region in the near-zero range and a sparse region with rare large values. The dense region contains most of the data, and the sparse region consists of values that significantly impact model accuracy. Therefore, accurately representing both regions is essential to minimize model performance degradation due to quantization.

In this experiment, we analyze the performance of piecewise bit allocation in both dense and sparse regions. The effect depends on the location and number of boundaries that divide regions, which means when setting the boundaries, it is crucial to accurately ensure the granularity required by each region to minimize accuracy degradation in the model. Therefore, in this experiment, we set four boundaries at 10%, 20%, 40%, and 60% for 2-region piecewise bit allocation and measure the MSE in both dense and sparse regions for each setting. In the 4-region piecewise bit allocation experiment, we segmented the data into four distinct regions. These segmentations are determined by the three boundaries that demonstrated the lowest MSE in the previous 2-region quantization experiment. The models used in this experiment are Vision Transformer, Stable Diffusion v2, and Llama 2, and the dense and sparse regions are divided into 99% and 1% of the total data region, respectively.

To perform in-situ quantization and evaluate the accuracy of quantization process, we set a specific batch size for each model and obtain the maximum value within a batch to determine a scaling bias. By using the scaling bias, quantization is performed independently on each tensor within every batch. Subsequently, we calculate the MSE between the original and quantized tensors within the dense regions and the sparse regions.

As shown in Figure 9, achieving low MSE values in both the sparse region and the dense region is challenging in the case of 2-region piecewise bit allocation. On the other hand, 4-region bit allocation exhibits relatively consistent MSE values but is higher in both of the regions. In the case of n-bit piecewise bit allocation with $N$-regions, the data bit-width allocated to each region is fixed to $n - log_2(N) - 1$. This means that even if the number of regions is increased to represent the entire data distribution more finely, the granularity for each region is suboptimal. This issue becomes a hurdle in accurately representing the entire region, including both sparse and dense regions. Consequently, this inflexible way of allocating bits leads to lower the overall accuracy of the model.

In contrast, SuFP consistently shows the lowest MSE values across all models and data regions. This is because SuFP ensures optimized granularity for each data region. Specifically, as explained in Section 3, SuFP uses a variable encoding field and variable data field to represent the data distribution, allowing for sufficient granularity for the high-granularity-required dense region while maintaining extensive dynamic

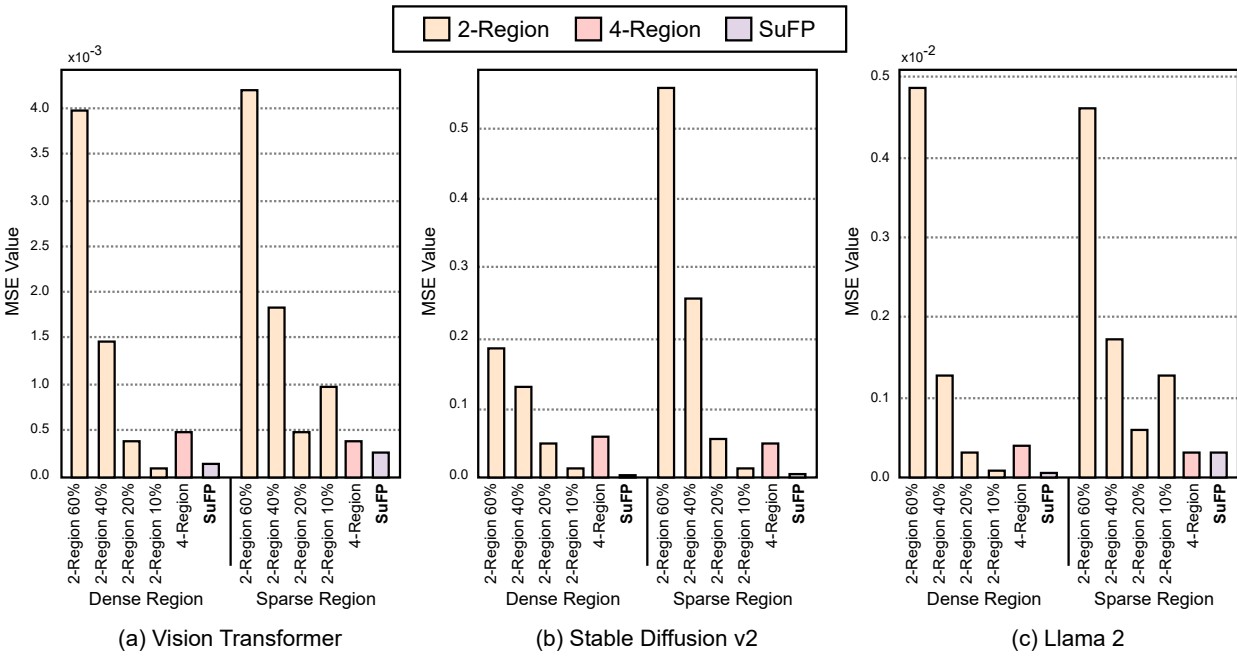

Figure 9: MSE value comparison of SuFP and other piecewise quantization techniques.

range for the sparse region demanding wide coverage. This approach effectively represents the real data distribution across each region, minimizing degradation in model performance.

## Appendix B  Analysis of SuFP Representations' Impact on SuFP PE Bit Utilization

In this session, we analyze the impact of three different representations of SuFP on bit utilization in SuFP PE. Each representation has varied mantissa and exponent bit-widths, enabling optimized bit allocation in both dense and sparse regions. Representation ① has the largest mantissa bit-width, while representation ③ has the smallest. These differences also affect the design of SuFP PE ALU. The bit-width of SuFP PE ALU is set based on representation ①, which has the largest mantissa bit-width. Although representation

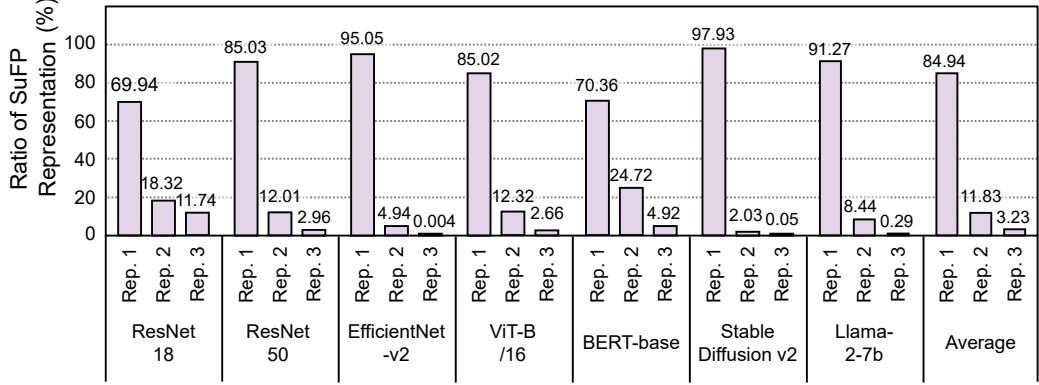

Figure 10: Comparison of SuFP representation proportion in weight and activation tensors across various models.

③ is primarily utilized to effectively represent outliers, it results in relatively lower bit utilization within SuFP PE.

Figure 10 shows the ratio of the three representations when applying SuFP quantization across various models. As shown in Figure 10, the ratio of representation ③ is noticeably low in all models. For instance, in EfficientNet-v2, representation ③ accounts for only about 0.004%. In conclusion, while representation ③ is essential for model accuracy, its low representation ratio indicates a minor impact on SuFP PE hardware's bit utilization. Consequently, SuFP demonstrates its effectiveness in terms of both accuracy and hardware's bit utilization.

## Appendix C    Implementation Details of Quantization Flow

---

**Algorithm 2:** scaling bias-optimal quantization flow

---

**Input** : Model $M$, Bias searching range $[b_i, b_j]$,
            Full-precision weight $W_{fp}$, Full-precision Activation $A_{fp}$
**Output:** Optimal scaling weight bias set $b_{w\_opt}[0 \ldots n-1]$,
            Optimal scaling activation bias set $b_{a\_opt}[0 \ldots n-1]$

1   // Optimize weight bias set
2   **for** $layer \leftarrow 0$ *to* $n-1$ **do**
3     $accuracy_{max} \leftarrow 0$
4     $W_{quant}[0 \ldots layer-1] \leftarrow Quant(W_{fp}[0 \ldots layer-1], b_{w\_opt}[0 \ldots layer-1])$
5     **for** $bias = b_i$ *to* $b_j$ **do**
6       $W_{quant}[layer] \leftarrow Quant(W_{fp}[layer], bias)$
7       $W' \leftarrow \{W_{quant}[0 \ldots layer], W_{fp}[layer+1 \ldots n-1]\}$
8       $accuracy \leftarrow Test(M, W', A_{fp})$
9       **if** $accuracy > accuracy_{max}$ **then**
10         $accuracy_{max} \leftarrow accuracy$
11         $b_{w\_opt}[layer] \leftarrow bias$

12   // Optimize activation bias set
13   **for** $layer \leftarrow 0$ *to* $n-1$ **do**
14     $accuracy_{max} \leftarrow 0$
15     $A_{quant}[0 \ldots layer-1] \leftarrow Quant(A_{fp}[0 \ldots layer-1], b_{a\_opt}[0 \ldots layer-1])$
16     **for** $bias \leftarrow b_i$ *to* $b_j$ **do**
17       $A_{quant}[layer] \leftarrow Quant(A_{fp}[layer], bias)$
18       $A' \leftarrow \{A_{quant}[0 \ldots layer], A_{fp}[layer+1 \ldots n-1]\}$
19       $accuracy \leftarrow Test(M, W_{quant}, A')$
20       **if** $accuracy > accuracy_{max}$ **then**
21         $accuracy_{max} \leftarrow accuracy$
22         $b_{a\_opt}[layer] \leftarrow bias$

23   **return** $b_{w\_opt}, b_{a\_opt}$

---

The scaling bias of SuFP is set as a single value within each layer. In addition, once the bias is determined, it remains invariant throughout the entire inference process of the model. The bias determines the quantization range and precision, significantly impacting the overall accuracy. Thus, employing a proper bias for optimization is extremely important.

In seeking the tensor-wise optimal bias, there are potential risks of slipping into local optimization instead of achieving global optimization. Concurrently, individual optimal bias for weight and activation might not yield the best outcome when computed together.

Based on these insights, we structure our optimization process as described in Algorithm 2. This process considers (i) interactions among adjacent layers, (ii) cumulative influences across the layers, and (iii) the synergistic relationship between weight and activation. Additionally, the total time required for this procedure depends on the target accuracy level, which can dynamically change the bias searching range.

As additional implementation details for the training experiments described in Section 4.4, we trained the classifier for 100 epochs using an SGD optimizer with a learning rate of 0.1, momentum of 0.9, and weight decay of 5e-4. We applied loss scaling (Mellempudi et al., 2019), which allows small gradient values to be represented with a smaller bit-width representation. To dynamically find the scaling bias for tensor quanti-

zation, we set bias as $bias = floor(log2(max(abs(x)))) - margin$. For weight and activation quantization, we set the *margin* term to 4 for the CIFAR-10 dataset and 2 for the CIFAR-100 dataset. For gradient quantization, we set the term to 1 for both datasets. We do not quantize the input of the first convolution and final fully connected layer to stabilize the training procedure (Mellempudi et al., 2019). We used modified ResNet-18 architecture for CIFAR-10/100 datasets; the first convolution layer is replaced by kernel size 3 and stride 1, and the last pooling layer is replaced by the identity function.

## Appendix D   QSNR results between SuFP and existing formats

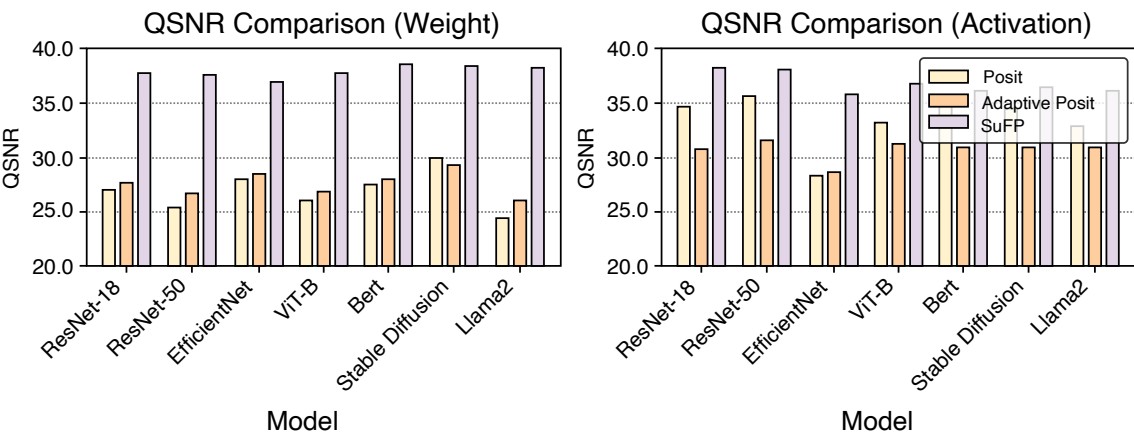

Figure 11: QNSR analysis of SuFP and Posit data types across various models

Posit covers a wider range of values, including those extremely close to zero. This allows Posit to handle a broader spectrum of numerical values effectively. On the other hand, SuFP is designed to cover a relatively narrower range of data, focusing on higher precision near zero but not at the extreme near-zero values. SuFP does not represent values extremely close to zero as effectively as Posit, but it allocates more bits to the data near zero, thereby achieving higher precision for these values.

It is important to note that these extremely near-zero values generally have minimal impact on the results after operations such as GEMM (General Matrix Multiply). In neural network computations, allocating more bits to significant values rather than extremely near-zero values often results in improved accuracy. This is because the more meaningful values contribute more significantly to the overall computation.

To further substantiate this point, we used the QSNR metric (Quantization Signal-to-Noise Ratio), which is widely adopted in recent data type and quantization studies. QSNR provides a comprehensive measure of how well the quantized values preserve the original data distribution, which is crucial for maintaining model accuracy and performance. We compared the QSNR performance of SuFP, Posit, and Adaptive Posit across various models, including ViT, ResNet18, ResNet50, EfficientNet, Stable Diffusion, BERT, and Llama2. As shown in Figure 11, and they consistently show that SuFP achieves higher QSNR values. This indicates that SuFP maintains higher numerical accuracy and robustness across diverse tasks.

In conclusion, the specific allocation of these bits within an 8-bit data type, as trivial as it may seem, has significant implications for performance when applied to neural network models. SuFP is meticulously optimized for the data distributions found in neural networks, particularly targeting near-zero dense regions and sparse outliers. As previously mentioned, our QSNR measurements indicate that SuFP consistently outperforms Posit in representing these distributions, especially in large models and diverse tasks, demonstrating superior accuracy.

Additionally, SuFP employs fixed encoding fields with corresponding fixed exponent/mantissa bit-widths, which simplifies its hardware implementation compared to Posit. Posit formats require dynamic detection of leading zeros and ones, leading to increased hardware overhead. For instance, Posit decoders, particularly

for es=2, can occupy roughly twice the area of a 4-bit multiplier, adding considerable hardware complexity. SuFP, on the other hand, maintains a simpler decoder design while achieving high performance across various models and tasks.

## Appendix E    Comparison with Block Floating-Point: Quantization Accuracy and Hardware Efficiency

We compare SuFP accuracy against standard formats like FP32 and BF16 as well as other PTQ techniques, including MSFP and BSFP. We use the configuration of MSFP and BSFP based on their reported superior accuracy in previous studies. Furthermore, we set the number of elements in a block to 16, as this value gives optimal performance for both MSFP and BSFP.

Table 7: Comparison of normalized accuracy among vision models with SuFP and other data types.

| Method | Data Type | Vision | | | |
| | Weight / Activation | ResNet-18 | ResNet-50 | EfficientNet-v2 (s) | ViT-B/16 |
|---|---|---|---|---|---|
| FP32 | FP32 / FP32 | 1.0000 (69.76/69.76) | 1.0000 (76.13/76.13) | 1.0000 (81.31/81.31) | 1.0000 (81.07/81.07) |
| BF16 | BF16 / BF16 | 1.0006 (69.80/69.76) | 0.9997 (76.11/76.13) | 1.0000 (81.31/81.31) | 0.9996 (81.04/81.07) |
| MSFP[1] | MSFP / MSFP | 0.9990 (69.69/69.76) | 0.9991 (76.06/76.13) | 0.9983 (84.09/84.23) | 0.9993 (81.01/81.07) |
| BSFP[2] | BSFP / MSFP | 0.9987 (69.67/69.76) | 0.9993 (76.08/76.13) | 0.9980 (84.06/84.23) | 0.9981 (80.92/81.07) |
| **SuFP** | **SuFP / SuFP** | **1.0007 (69.81/69.76)** | **1.0005 (76.17/76.13)** | **0.9994 (81.26/81.31)** | **0.9999 (81.06/81.07)** |

[1] The precision of MSFP is characterized as MSFP16 (1bit sign, 7bit mantissa, 8bit exponent).
[2] BSFP is structured with 5-bit and 2-bit mantissa, accompanied by 8-bit and 7-bit scale factors corresponding to each mantissa.

**Evaluation on Vision Models.** Table 7 compares the performance of various quantization techniques in the vision domain. The top-1 accuracy metric is used for performance evaluation. For a consistent comparison, we source the accuracy results for MSFP and BSFP from the BSFP paper. To ensure consistency, we set up our environment on FP32 accuracy from the BSFP paper. However, FP32 accuracy for EfficientNet-v2 differs from the reported value. Thus, we normalize the accuracies of MSFP, BSFP, and SuFP relative to FP32 and focus on comparing their changes.

Table 8: Comparison of performance among language and text-to-image generative models with SuFP and other data types.

| Method | Data Type | BERT-base | | | Stable Diffusion v2 |
| | Weight / Activation | MRPC ↑ (Accuracy) | CoLA ↑ (MCC) | SQuAD 2.0 ↑ (F1-score) | COCO ↓ (FID-score) |
|---|---|---|---|---|---|
| FP32 | FP32 / FP32 | 0.8307 | 0.5678 | 78.8684 | 27.0643 |
| MSFP | MSFP / MSFP | 0.8319 | 0.5636 | 78.8113 | 27.2551 |
| BSFP | BSFP / MSFP | 0.8336 | 0.5636 | 78.7647 | - |
| **SuFP** | **SuFP / SuFP** | **0.8371** | **0.5866** | **78.9547** | **25.6262** |

**Evaluation on Language and Text-to-Image Models.** Table 8 shows the performance of SuFP on the Language and Text-to-image categories. For comparison, we use FP32, MSFP, and BSFP as baseline. In the Language category, BERT-base serves as our representative model. We evaluate the BERT-base model on the MRPC, CoLA, and SQuAD 2.0 datasets using accuracy, MCC, and F1-Score as the performance metrics, respectively.

For the Text-to-image category, we experiment with the Stable Diffusion v2 model. We use the COCO dataset in this experiment, adopting the FID score as our performance metric. For the experiment, we adopted our SuFP only to diffusion.

Table 9: Comparison of performance among LLMs with SuFP and other data types.

| Method | Data Type | Llama 2-7b | | | | |
| | Weight / Activation | STEM | Humanities | Social Sciences | Other | Average |
|---|---|---|---|---|---|---|
| MSFP | MSFP / MSFP | 0.372 | 0.431 | 0.523 | 0.524 | 0.460 |
| **SuFP** | **SuFP / SuFP** | 0.370 | 0.426 | 0.516 | 0.516 | 0.455 |

**Evaluation on LLMs.** Table 9 shows the performance of SuFP and MSFP on the LLMs. We evaluate the Llama 2 model on MMLU benchmark.

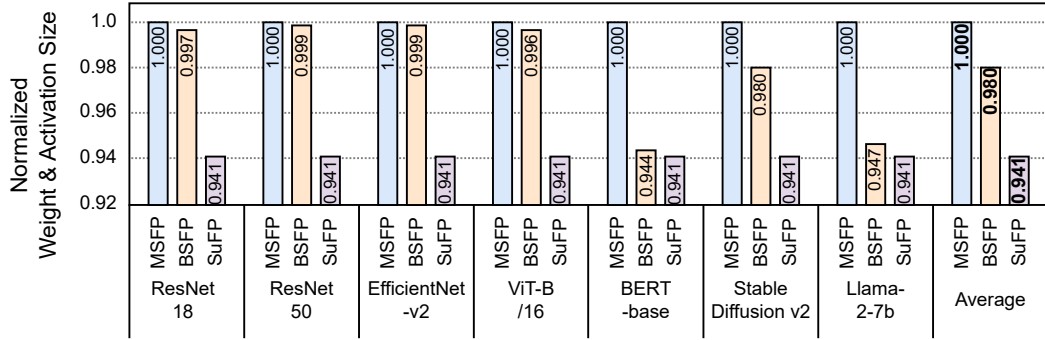

Figure 12: Memory footprint of MSFP, BSFP and SuFP normalized to MSFP.

**Memory Footprint** Figure 12 shows the required memory footprint for the weight and activation tensors of models applied with the quantization techniques of SuFP, MSFP, and BSFP. They are normalized with respect to MSFP to illustrate the effective reduction in footprint clearly. It is worth noting that BSFP uses 128-bit memory due to the standard byte alignment despite its 127-bit configuration, causing 0.8% overhead. Based on the calculations, SuFP occupies 0.941× of the memory of MSFP and 0.960× that of BSFP on average.

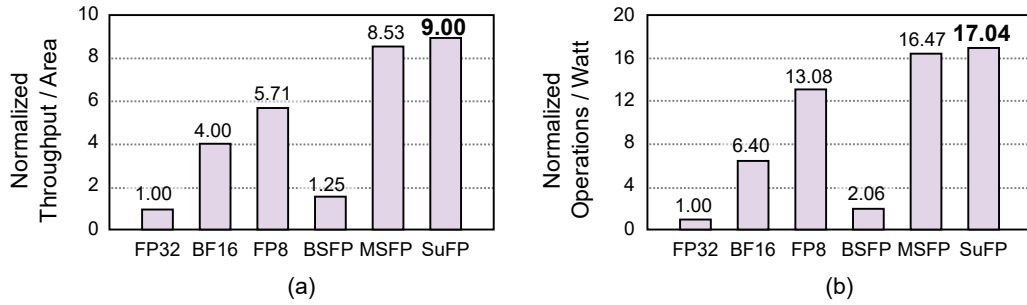

Figure 13: (a) Normalized throughput per area and (b) normalized operations per watt comparison of SuFP with other baselines. The values are normalized with respect to FP32.

**Hardware Efficiency** We compare the hardware efficiency of SuFP with that of other methods. Figure 13 (a) shows the throughput per area of PEs for various data formats, normalized with the result of FP32. The SuFP PE demonstrates the highest efficiency, 9.00× compared to FP32 PE. On the other hand, BSFP shows a lower value due to its use of bit-serial PE. The bit-serial operation in BSFP PE necessitates 16

computations to process the multiplication of a 7-bit BSFP mantissa with a 7-bit MSFP mantissa using a 2-bit multiplier. In the case of MSFP, while MSFP utilizes a 7-bit multiplier to multiply mantissa, SuFP employs a 6-bit multiplier, enabling SuFP PE to achieve enhanced throughput-per-area efficiency. In more detail, SuFP PE is up to $7.20\times$ more efficient than that of state-of-the-art MSFP and BSFP PE.

We also evaluate the performance of SuFP PE in terms of energy efficiency. Figure 13 (b) shows the comparison results, presenting the number of operations per watt, which is normalized to FP32. SuFP outperforms FP32 PE by being $17.04\times$ more energy-efficient. Even when compared to the state-of-the-art MSFP and BSFP, our PE is up to $8.27\times$ more energy-efficient.

## Appendix F   Implementation Details of Various PEs

In Section 4.2, we present an analysis of QSNR and hardware efficiency for SuFP and various FP8 configurations. For these experiments, we implemented SuFP PE and various baseline PEs (FP32 and various FP8 configurations) using SystemVerilog and synthesized them using the Synopsys Design Compiler in 28nm CMOS technology. All implementations include pipeline flip-flops inserted to achieve a target frequency of 1 GHz. By using this setup, we measured the area and power values of the PEs. For a clearer comparison, the results are presented in Table 10 with the same throughput value for each PE. To provide further understanding of these results, the detailed configurations for each PE are described as follows:

- **FP32 PE** supports full-precision FP32 operations, involving multiplication and accumulation operations in FP32 format for precise outcomes.

- **FP8 PE** is configured with different variations including E2M5, E3M4, E4M3, and E5M2. Each configuration performs multiplication operations in the specified FP8 format, with a BF16 format accumulator used to ensure consistent accuracy.

- **SuFP PE** conducts 16 SuFP multiplication operations in parallel, as shown in Figure 4. SuFP PE also utilizes a BF16 accumulator, similar to the other PEs.

Table 10: Iso-throughput area and power of SuFP and other baselines.

|  | 16x FP32 | 16x FP8(E2M5) | 16x FP8(E3M4) | 16x FP8(E4M3) | 16x FP8(E5M2) | SuFP |
|---|---|---|---|---|---|---|
| Area ($\mu m^2$) | 29731.9679 | 5844.3840 | 5628.6720 | 5205.3120 | 4806.1440 | 3303.7200 |
| Power ($mW$) | 19.8528 | 1.9920 | 1.8080 | 1.5173 | 1.0966 | 1.1649 |

In addition, to further clarify the hardware composition, we analyzed the area contributions of each constituent module in the proposed SuFP PE. Specifically, the SuFP ALUs account for 36.8% of the total area, the mantissa alignment for 28.8%, the adder tree for 23.3%, and the accumulator for 10.9%, respectively.

## Appendix G   Extension of SuFP PE to Systolic Array

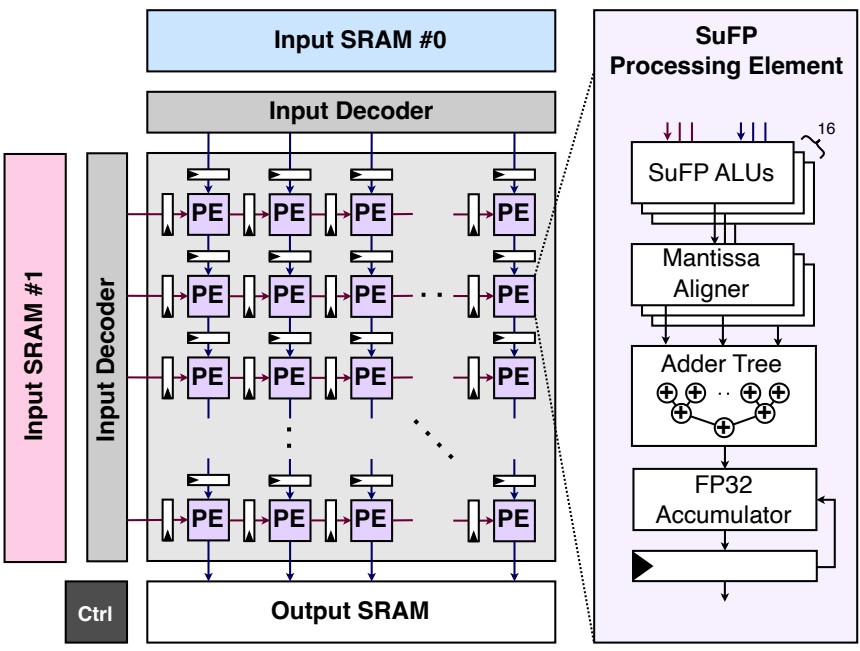

Figure 14: Systolic array architecture containing proposed SuFP PEs.

A systolic array architecture consists of multiple PEs arranged in a 2D array format, allowing parallel data processing. This architecture is not only adopted by Google's TPU (Jouppi et al., 2021) but is also widely utilized across various NPUs such as Venkataramani et al. (2021); Geva et al. (2022); Gomes et al. (2022). Our proposed SuFP PE can also seamlessly integrate into the systolic array architecture, potentially leading to significant performance enhancements. Figure 14 illustrates the SuFP PE (on the right) and the systolic array architecture composed of these SuFP PEs (on the left). The PEs adjacent to the SRAM directly receive the decoded data from the SRAM. Subsequently, these PEs use this data for computations and transmit the computed results and input data to neighboring PEs. Through this process, once-decoded data efficiently propagates among PEs, thus minimizing the decoding-related overhead in the systolic array.

# Appendix H    Additional Quantization Results on Text-to-Image Generation

In this section, we provide the results of text-to-image generation using SuFP quantization applied to full-precision diffusion models. As shown in Figure 15, there is almost no difference between images generated with full-precision and those produced using SuFP quantization.

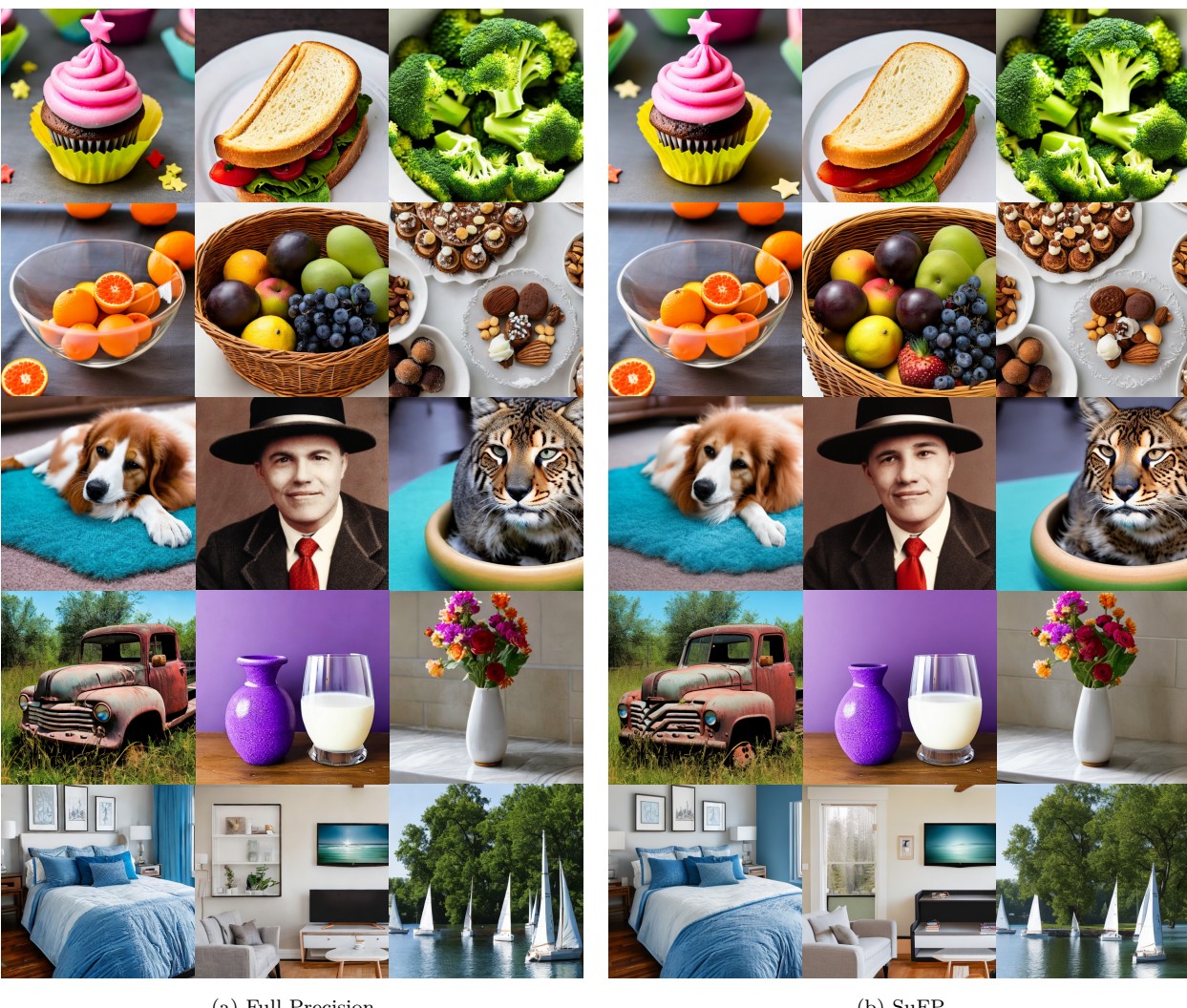

(a) Full-Precision                          (b) SuFP

Figure 15: Sample images generated from Stable Diffusion model on COCO dataset with full-precision and our SuFP.

