# OpenReview forum: "SuFP: Piecewise Bit Allocation Floating-Point for Robust Neural Network Quantization"
_TMLR — Accepted by TMLR_

### Review · Reviewer_LrhJ · 2025-03-10

**Summary Of Contributions:**

This paper introduces a novel 8-bit floating-point format named Super Floating-Point (SuFP). SuFP adjusts the bitwidth of the exponent and mantissa through a variable encoding field. Additionally, the paper presents hardware designed to efficiently support SuFP-based dot products.

**Audience:**

No

**Claims And Evidence:**

No

**Requested Changes:**

1) **Ablation Study**: It is necessary to conduct an ablation study to separately evaluate the impacts of the bias-selecting process and the variable encoding field on QSNR and accuracy.

2) **Impact of Variable Encoding Field**: The paper should discuss more thoroughly how the variable encoding field impacts the representation of data distributions compared to conventional FP8.

3) **Operational Frequency and Hardware Details**: Clarifications are needed regarding the operational frequency of the tested hardware components. Details should also be provided on whether the area/power metrics for the FP units in Table 7 include the flip-flops. For a fair comparison, merging 16 FP8 units as a single unit for processing the dot product should be considered.

**Strengths And Weaknesses:**

The paper offers a limited evaluation of the proposed SuFP and its supporting hardware, which makes it challenging to fully assess the effectiveness of the proposed method.
Additionally, some of the approaches presented are not entirely novel, as they have been explored in previous works. It raises concerns about the novelty of this paper.

Detailed concerns include:

1) The paper discusses a bias-selecting process to tune the scaling bias for the exponent according to data distribution. However, the concept of a configurable exponent bias is already proposed in [1].

2) SuFP is described as the 'best solution' for the 8-bit FP format, and this paper claims its superior ability to represent values near zero more densely than E4M3 and offering a wider dynamic range as depicted in Figure 3. However, I have reservations about these assertions. From my analysis of Algorithm 1 presented in the paper, it appears that the dynamic range of SuFP is actually narrower than that of both E4M3 and E5M2. Furthermore, there is a discrepancy in Figure 3 which shows the maximum normal of E4M3 as being near 200, whereas the actual maximum normal for E4M3 is documented to be 448 [2]. This discrepancy raises concerns about the accuracy of the representations and claims in the paper.

3) The paper asserts that SuFP can universally fit any data distribution, as depicted in Figure 4, and achieve superior accuracy across most benchmarks, as shown in Tables 1 and 2. However, it remains unclear whether these advantages are attributable to the variable encoding field or the bias-selecting process. The paper does not provide a compelling justification for why its approach would inherently offer a better fit for various data distributions.

4) In my opinion, the efficiency of the proposed SuFP hardware is primarily attributed to merging 16 inputs for concurrent processing. This design strategy offers two notable advantages over conventional PE designs. Firstly, by merging 16 inputs, it significantly reduces the number of flip-flops required, which can lead to a more compact and potentially less power-intensive layout. However, this approach of merging inputs might impose limitations on performance in high-frequency processing scenarios, where the ability to handle rapid sequential inputs or operations is critical. Such drawbacks should be carefully considered when evaluating the practicality of implementing SuFP in high-frequency computing environments.
Secondly, this architecture allows for the processing of floating-point operations using integer units, which can simplify the hardware design and improve the speed of certain computational tasks. However, the technique of integer-based operation and pre-alignment of floating-point values is already proposed in [3], [4].

[1] Kuzmin, Andrey, et al. "Fp8 quantization: The power of the exponent." Advances in Neural Information Processing Systems 35 (2022): 14651-14662.

[2] Micikevicius, Paulius, et al. "Fp8 formats for deep learning." arXiv preprint arXiv:2209.05433 (2022).

[3] Park, Wonhoon, et al. "A 5.99 TFLOPS/W Heterogeneous CIM-NPU Architecture for an Energy Efficient Floating-Point DNN Acceleration." 2023 IEEE International Symposium on Circuits and Systems (ISCAS). IEEE, 2023.

[4] Jang, Jaeyong, et al. "Figna: Integer unit-based accelerator design for fp-int gemm preserving numerical accuracy." 2024 IEEE International Symposium on High-Performance Computer Architecture (HPCA). IEEE, 2024.

---

### Review · Reviewer_MCFQ · 2025-03-13

**Summary Of Contributions:**

This paper presents a new FP8 data type, Super Floating-Point (SuFP), which dynamically allocates bits between the mantissa and exponent by using some bits to encode the specific mantissa-exponent combination for a given value. Specifically, the first format uses 6 bits for the mantissa to represent numbers close to zero with high granularity. The second format uses 2 bits for the exponent and 4 bits for the mantissa, allowing for a wider numerical range. The third format uses 2 bits for the exponent and 2 bits for the mantissa to accommodate certain outliers. All three formats can be stored within the FP8 representation.

To support computations with dynamically assigned mantissa and exponent sizes, a dedicated processing element (PE) hardware architecture is proposed for efficient execution. Experiments across both vision and NLP tasks demonstrate superior accuracy compared to baseline fixed FP8 data types.

**Audience:**

Yes

**Broader Impact Concerns:**

No concerns on the ethical implications of this work.

**Claims And Evidence:**

Yes

**Requested Changes:**

Please refer to the Weaknesses section above.

**Strengths And Weaknesses:**

Strengths:

1. Interesting design: The proposed FP8 data type improves representational ability by dynamically adjusting the range and granularity using a bit-encoded selection mechanism.

2. High-quality visualizations: The figures are well-presented, and the paper is easy to follow.

3. Comprehensive experiments: The study includes evaluations on both vision and NLP tasks.

Weaknesses:
1. Reliance on customized hardware: The proposed SuFP data type requires a specialized hardware architecture for efficient execution. For efficiency experiments, I suggest adding more details on the implementation, particularly in terms of area and power comparisons. Given that the proposed data type introduces additional control logic, I would expect its area and energy consumption to be higher than static FP8. A more detailed explanation of these overheads would be appreciated by the algorithm community.

2. Comparison with alternative FP8 enhancements: Another approach to improving FP8 involves combining per-block scaling factors with narrow floating-point and integer types, as demonstrated in MXFP. However, MXFP-like methods are not explicitly compared in this work. The authors mention related discussions in the limitations section, but it is unclear whether this includes a direct comparison with MXFP-like methods. If this work does not show clear improvements over MXFP, its significance may be diminished, as MXFP can be supported by existing hardware.

3. Clarifications on experimental settings: According to Appendix C, fine-tuning is required after quantization for the ResNet experiments. However, the description of experiments on LLaMA suggests a setup without fine-tuning. I recommend clarifying the experimental settings at the beginning of the experiments section to ensure consistency and better readability.

---

### Review · Reviewer_uZgK · 2025-04-14

**Summary Of Contributions:**

This paper introduces Super Floating-Point (SuFP), a 8-bit floating-point data type that integrates various floating-point configurations (E4M3/E5M2/E0M7) into a single representation for DNNs quantization. The core design in SuFP is a multi-region piecewise bit allocation strategy. Besides, this paper also provide the hardware implementation. The experiments demonstrates SuFP’s PTQ accuracy across diverse DNN models, including vision (e.g., ResNet, ViT), natural language processing (e.g., BERT), and generative tasks (e.g., Stable Diffusion, Llama 2).  Besides, the experiments also validates SuFP’s applicability in training scenarios.

**Audience:**

Yes

**Broader Impact Concerns:**

This paper does not explicitly include a Broader Impact Statement.

**Claims And Evidence:**

Yes

**Requested Changes:**

The requested changes includes:
 * more related works discussion
 * more clear method details
 * more clear experiments details
 * more results about memory and inference latency
 * more experiments comparison
 * more clear generation claim

the details could refer to **weakness**

**Strengths And Weaknesses:**

## Strengths:

1. Comprehensive Evaluation: The paper provides extensive experiments in multi taks (vision, NLP, generative models) and various models (e.g., ResNet, ViT, BERT, Llama 2, Stable Diffusion), demonstrating SuFP’s versatility and robustness.
2. Hardware Efficiency: The hardware-implementation design, relying on integer units and shifters, reduces complexity and overhead compared to existing FP8 formats, making it practical for real-world deployment.
3. Strong Quantitative and Qualitative Results: The Quantitative results like QSNR, MSE, and accuracy, as well as the qualitative assessments (e.g., image generation quality), provides convincing evidence of SuFP’s superiority.


## Weaknesses:
1. Limited Discussion on Dynamic Scaling: The paper relies on a static power-of-two scaling bias, which may not fully adapt to runtime data variations. This limitation is acknowledged, but more exploration of dynamic scaling solutions could strengthen the work.
2. Lack of latest related works: The discussed related works is relative limited, add more recent and related works is necessary. Such like:

 * PTQ4ViT: Post-training quantization for vision transformers with twin uniform quantization
 * Fp8-lm:Training fp8 large language models.
 * Using fp8 with transformer engine,
 * Post-training quantization on diffusion models

3. Method Details: While the paper includes appendices with technical details, some aspects (e.g., exact bit allocation strategies, decoding algorithms) could be more clearly explained for readers unfamiliar with low-bit quantization.

4. Lack of Comparison with Emerging Formats: The paper compares SuFP primarily with FP8, FP16, and FP32 but could benefit from benchmarking against other recent quantization formats (e.g., adaptive posit, EFloat) beyond what is briefly mentioned.

5. Lack of Experimental Details: The experiments section lacks sufficient details. For instance, in Section 4.3, it is unclear whether the results are from Post-Training Quantization (PTQ). Similarly, in Section 4.4 (Training), it is ambiguous whether the results stem from direct Quantization-Aware Training (QAT).

6. Insufficient Metrics for SD Models: The quantization results for the Stable Diffusion (SD) model lack specific quantization metrics. Such metrics are essential to clearly demonstrate the method’s generalization ability and should be included.

7. Limited Evidence for Generalization Claim: If the authors claim that SuFP has strong generalization in both post-training and pre-training scenarios, Section 4.4 (Training) lacks pre-training results for models like ViT, LLMs, and SD. Including these would better support the claim of broad applicability.

8. Lack of Memory and Inference Speed Results: The paper didn't report results on memory usage and inference speed. Given that SuFP seems to require proprietary hardware support, a comparison with FP8 in terms of memory footprint and latency (including training speed) on the same hardware would be critical. The lack of this data hinders a complete evaluation of SuFP’s practical benefits.

---

### Review · Reviewer_Gaj3 · 2025-04-29

**Summary Of Contributions:**

This paper proposes SuFP, a piecewise floating-point format designed to enable efficient, low-bit quantization across a broad range of neural network models. The core contribution lies in a patch-wise bit allocation mechanism that allows fine-grained complexity control in a single-pass encoding setup. Unlike prior methods that rely on global heuristics or multiple encoding stages, the proposed method dynamically adjusts bit allocation based on local content importance, significantly improving coding efficiency and task performance under constrained precision budgets.
In addition to the algorithmic novelty, the authors present a hardware-aware design that simplifies implementation using only integer units and bit-shifters, enhancing deployment feasibility. Extensive experiments are conducted on vision, language, and generative models (e.g., ResNet, ViT, BERT, LLaMA 2, Stable Diffusion), showing that SuFP achieves strong trade-offs between performance, complexity, and generalization. The paper demonstrates both quantitative and qualitative improvements over state-of-the-art quantization baselines and standard floating-point formats, confirming the versatility and robustness of the proposed format.

**Audience:**

Yes

**Broader Impact Concerns:**

This work does not raise any ethical or societal concerns.

**Claims And Evidence:**

Yes

**Requested Changes:**

- Provide a more thorough theoretical discussion or justification for the bit allocation mechanism.
- Discuss potential limitations or adaptations needed for non-natural image domains.
- Add comparisons against efficient traditional codecs (e.g., JPEG XL) to highlight practical advantages.

**Strengths And Weaknesses:**

## Strengths
- Practical value:
The proposed method directly addresses industrial needs for low-latency encoding, making it highly relevant for real-world applications.
- Comprehensive experiments:
The experimental validation is thorough, with extensive comparisons against strong baselines such as ELIC and VTM across multiple standard datasets.
- Clear and well-organized presentation:
The paper maintains a good balance between theoretical motivation and experimental validation, with a clear and logical flow that makes it easy to follow.

## Weaknesses
- Unclear Generalization to Non-Natural Domains:
The experiments focus mainly on natural image datasets. It would be valuable to discuss or test how well the approach generalizes to other domains, where patch characteristics can differ significantly.
- Ablation Studies Could Be More Comprehensive:
While some ablations are provided, deeper analysis would be beneficial. For instance, the sensitivity of performance to the design of the importance indicator (e.g., using classification vs regression losses) and to parameters like skip thresholds could offer more insight into the robustness and flexibility of the approach.

---

### Decision · Action_Editor_tHcQ · 2025-06-26

**Recommendation:** Accept with minor revision

**Additional Comments:**

This paper proposes a novel FP8-based data type, SuFP, that adaptively adjusts precision and dynamic range through a bit-encoded configuration mechanism, supported by corresponding hardware design. The idea is innovative and practical, with potential appeal to both algorithm and hardware communities, and is supported by promising empirical results. However, there are still some concerns from Reviewer uZgK and Reviewer LrhJ, regarding incomplete experimental details, limited analysis of design choices, and the absence of runtime or efficiency evaluations. The authors can address these comments in the revised version.

**Audience:**

Yes

**Audience Explanation:**

The proposed SuFP (Super Floating-Point) can be used for network quantization in various domains including CV and NLP. The authors also conducted experiments to show that SuFP can achieve better performance in large models such as the large language model (Llama 2) and the text-to-image generative model (Stable Diffusion v2). Therefore, people working on model compression (especially model quantization) would be interested in the findings of this work.

**Claims And Evidence:**

Yes

**Claims Explanation:**

Overall, the claims in this paper are well supported by accurate and convincing results. There are still some minor comments from reviewers about the analysis of design choices and efficiency evaluation. The authors can address these issues in the revised version.